# Role of Autophagy-Related Gene *atg22* in Developmental Process and Virulence of *Fusarium oxysporum*

**DOI:** 10.3390/genes10050365

**Published:** 2019-05-13

**Authors:** A. Rehman Khalid, Shumin Zhang, Xiumei Luo, Khalid Mehmood, Junaid Rahim, Hamayun Shaheen, Pan Dong, Dan Qiu, Maozhi Ren

**Affiliations:** 1School of Life Sciences, Chongqing University, Chongqing 401331, China; youran@cqu.edu.cn (S.Z.); aoamateen319@yahoo.com (X.L.); haseeb78692@yahoo.com (P.D.); 2Department of Plant Pathology, University of Poonch AJK, Rawalkot 12350, Pakistan; 3Department of Entomology, University of Poonch AJK, Rawalkot 12350, Pakistan; kmmaldial@yahoo.com (K.M.); junaidrahim47@yahoo.com (J.R.); 4Department of Botany, The University of Azad Jammu & Kashmir, Muzafarabad13100, Pakistan; hamayun@hotmail.com

**Keywords:** *Fusarium oxysporum*, autophagy, potato dry rot, filamentous fungi, virulence

## Abstract

Autophagy is a universal catabolic process preserved in eukaryotes from yeast to plants and mammals. The main purpose of autophagy is to degrade cytoplasmic materials within the lysosome/vacuole lumen and generate an internal nutrient pool that is recycled back to the cytosol during nutrient stress. Here, *Fusarium oxysporum* was utilized as a model organism, and we found that autophagy assumes an imperative job in affecting the morphology, development, improvement and pathogenicity of *F. oxysporum*. The search of autophagy pathway components from the *F. oxysporum* genome database recognized putative orthologs of 16 core autophagy-related (*ATG*) genes of yeast, which additionally incorporate the ubiquitin-like protein *atg22*. Present study elucidates the unreported role of *Foatg22* in formation of autophagosomes. The deletion mutant of *Foatg22* did not demonstrate positive monodansylcadaverine (MDC) staining, which exposed that *Foatg22* is required for autophagy in *F. oxysporum*. Moreover, the *∆Foatg22* strains exhibited a decrease in hyphal development and conidiation, and reduction in pathogenicity on potato tubers and leaves of potato plant. The hyphae of ∆*Foatg22* mutants were less dense when contrasted with wild-type (WT) and overexpression (OE) mutants. Our perceptions demonstrated that *Foatg22* might be a key regulator for the control of dry rot disease in tuber and root crops during postharvest stage.

## 1. Introduction

The fungus *Fusarium oxysporum* is the most common pathogen infection of tubers and roots [1], which causes *Fusarium* wilting at the planting stage and dry rot at the storage stage, impacting its nutritive and economic value [2,3]. Also, this fungus can cause severe reductions in crop yield, often estimated between 6% and 25% annually throughout the world. Significant yield losses of 60% are recorded worldwide in storage conditions [4,5,6]. *Fusarium oxysporum* can be seed-borne or soil-borne; it can behave as a vascular colonization pathogen, and act as a decomposer [7,8]. Dry rot disease can be identified by shrinking, shriveling, and lesions on the tuber/root, with brown to black characteristics in the internal part of tuber/root [7,9,10]. The pathogen penetrates the tissues such as roots, tubers, and leaves through wounds and then induces rot infection. The infected tissues become dark red by forming streaks which rise to ground level. With the passage of time, the old leaves become yellow or flaky and then detach from the plant [11]. It was demonstrated that, in *F. oxysporum*, during vegetative growth, autophagy controls the number of nuclei in the hyphal compartment and is also essential for virulence [12].

Autophagy is a coordinated procedure which is liable for the quick declension of vast parts of the cytoplasm in the vacuolar lumen [13,14]. On the other hand, the recycling of degradation products, such as amino acids, generated in autophagy toward the cytoplasm is critical to cell physiology; however, the molecular details of this process remain to be adequately determined. The fission yeast *Schizosaccharomyces pombe* contains a homolog of *Saccharomyces cerevisiaeatg22*p which is a vacuolar membrane protein involved in the breakdown of autophagic bodies during autophagy [15]. The deduced amino-acid sequence of *S. pombeatg22* shared 21% similarity and 42% identity with *S. cerevisiaeatg22*. SOSUI investigation (http://bp.nuap.nagoya-u.ac.jp/sosui/) proposed that *S. pombeATG22* has eleven transmembrane helices. In spite of the fact that *S. pombeatg22* is essential for germination of spores [13], the criticalness of *atg22* in the cell compartmentalization of amino acids remains to be explored. Autophagy plays a fundamental role in cell differentiation and the development of pathogenicity in phytopathogenic fungi [16,17,18,19,20]. In rice blast fungus, *Magnaporthe oryzae,* the analysis of autophagy-related genes demonstrated that mutants lacking any of these genes failed to cause disease. Loss of any of the *atg* genes, except *Fgatg17*, prevented the fungus from causing *Fusarium* head blight disease in wheat [21]. The autophagy affects the formation of appressorium, which directly contributes to the development of diseases caused by fungus *Magnaporthe oryzae* [22,23,24]. Autophagy also affects the virulence of *Ustilago maydis*, the cause of corn smut, and penetrates the host through appressorium [20]. Studies reported that different autophagy-related genes were involved in the pathogenicity of *Verticillium dahlia*, including *VdPKAC1*, *VGB*, *VMK1*, *VGB*, *VdSNF1*, and *VdPKAC1* [25,26,27,28]. Here, we explored the function of autophagy-related gene *atg22* in the development and pathogenicity of *F. oxysporum*. We generated gene deletion mutant *∆atg22* by adopting a target gene replacement technique and generated an overexpressed mutant of *atg22*. The role of these mutants in the development of aerial hyphae, conidial formation, and pathogenicity was evaluated.

## 2. Materials and Methods

### 2.1. Isolation of Fungus and Culture Conditions

The strains of *Fusarium oxysporum* f. sp. *lycopersici* were isolated from diseased tubers of potato, and their identity was confirmed by plant infection assay before being used as wild-type (*WT*) strains in the present study. These strains were sensitive to HygB when used at a concentration greater than 30 mg/mL. A conidial suspension of strains was prepared and then stored at −80 °C with the addition of 50% glycerol. For further use, the conidia were grown again on fresh potato dextrose agar(PDA)medium at 25 °C For the preparation of mycelium and conidia, PDA liquid medium was used. The *A. tumefaciens* strains of agrobacteria GV3101 were used for the transformation of conidia of *F. oxysporum*. The strains of agro bacterium *A. tumefaciens* were grown on LB media.

### 2.2. Generation of Deletion Mutants

The *Foatg22* deletion mutant was generated with fusion PCR [29]. The *Foatg22* deletion mutant was created using HygB resistance cassette with the replacement of the open reading frame (ORF). The first 1000-bp upstream fragments, the HygB resistance cassette of 1040 bp from psilent-1 vector, the HygB resistance cassette from psilent-1 vector, and the downstream fragment of 1000-bp *Foatg22* were amplified with three pairs of primers, i.e., P1/P2, P3/P4, and P5/P6, respectively. After that, three fragments were fused with upstream−*Hph*−downstream, and then restriction enzyme digestion was performed with enzymes *AsiS*I and *Sbf*Iby ligation with vector PPk2 (Appendix A). The final recombinant plasmid of PPk2−U−*Hph*−D was transferred into WT according to a previously reported method [30]. The PDA media which was supplemented with 50 µM F2du (5-fluoro-2’-deoxyuridine) and 50 mg/mL HygB was used for screening of transformants. The transformants were confirmed through PCR screening with F-*hph*/R-*hph*. Then, qRT-PCR was used for identification of the deletion mutant followed by PCR screening. For identification of whether HygB was a single copy of the inserted mutant, we used F-*hph*/R-*hph* to 1021 bp.

### 2.3. Overexpression of Foatg22 Mutant Strains

The complementary DNA (cDNA) of *F. oxysporum* encompassed the complete *Foatg22* ORF and it was amplified with total RNA using primers atg22F/R, which corresponded to seven initial codons of ORF, restriction site *Not*I, and additional cytosine. For reverse complementation, *atg22*-12 corresponded to the last seven codons of the restriction site *Sbf*I, as well as additional cytosine. Then, the resulting band was cloned into the vector p8GWN. This plasmid was used as a template for amplification of PCR. The primer *atg22*-F corresponded to the first eight codons with *Sbf*I and *Asc*I restriction sites (Appendix A). The amplified fragment of DNA was cloned into the F303 vector, resulting in plasmid *Foatg22*/F303. A 2.8-kb fragment of *Foatg22* fusion under the control of the *trpC* promoter and the *trpC* terminator was obtained by PCR amplification using primer pair M13F and M13R, as mentioned in Appendix A, and it was later used for protoplast strain formation (Appendix A).

### 2.4. Fungal Transformation

According to a previously described method, the agro bacterium tumefaciens -mediated transformation (ATMT) process was performed with some modifications [30]. The strains of *A. tumefaciens* GV3101 containing PPk2 vector were grown on media amended with PDA at 28 °C and allowed to grow. When the optical density of strains at 660 nm (OD_660_) reached 0.5, the culture was mixed in conidial suspension (10^7^ conidia/mL) with equal concentration and then diluted to OD_600_ = 0.15 in the induction medium (IM) which contained 200 mM acetosyringone (AS), before being cultured on an orbital shaker at 200 rpm for 6 h at 28°C. Then, 250 mM of the mixture was kept on nitrocellulose filters (diameter, 80 mm, pore size, 0.45 mm; Whatman, Tokyo, Japan) for 48 h on co-cultivation medium. These filters were further transferred to a selective medium which was amended with cefotoxime (500 mg/mL) and HygB (50 mg/mL) to defeat cells of *A. tumefaciens*. The transformants were selected randomly after seven days and then transferred on PDA medium which was enriched with HygB (50 mg/mL).

### 2.5. Evaluation of Radial Growth, Conidiation, Formation, and Germination

In the present study, to evaluate conidial formation and radial growth of strains, the PDA medium was used which contained 50 mg/mL HygB. From a 10-day-old culture, the conidia of fungus was harvested and then filtered from lens paper containing two layers, before being re suspended with the concentration of 1 × 10^7^ spores/mL in sterile water. The 5-µL conidial suspension of *WT*, *Foatg22∆*, and overexpression strains was inoculated in each flask and plate. The plates were incubated at 25 °C. The diameter of the colony and the color of strains were observed and measured every day. To observe the conidial germination, in 1 mL of potato dextrose agar (PDB) medium, 10^2^ conidia were inoculated with continuous shaking at 150 rpm, and conidial germination was observed at 7, 12, 21, 28, 36, and 48 h. At 12 h, using a blood counting chamber, the germination rate of conidia was calculated [31]. To analyze the data, the software SPSS 15.0 for Windows^R^ (LEAD Technologies, Inc., Charlotte, NC, USA) was used. For statistical analysis, the Duncan post hoc test was used to observe the differences among strains *atp* ≤ 0.05. For calcofluor white staining, 5 × 10^7^ freshly obtain micro conidia were grown at 28 °C for 14h in 5 mL of PDA with shaking at 170 rpm, and incubated in the dark for five minutes with 10 µm of calcofluor white (CFW).

### 2.6. Analysis of Autophagy

In strains of *F. oxysporum*, to visualize the autophagy, 2.5 × 10^8^ micro conidia of different strains were inoculated on PDA medium and allowed to grow at 28 °C for 15 h. The strains were washed with sterile water and then transferred into synthetic medium (SM) medium lacking a nitrogen source in the presence or absence of 4 mM phenylmethylsulfonyl fluoride (PMSF) (Sigma, P7626). After starvation for 1h, the mycelium of strains was stained with monodansylcadaverine (MDC) (Sigma, D4008), at a concentration of 50 mM in the dark for 30 min. Then, strains were washed with water and observed under differential interference contrast (DIC) and epifluorescence microscopy.

### 2.7. Analysis of Gene Expression

The RNA of *OEFoatg22* and*∆Foatg22* mutants and WT of *F. oxysporum* was extracted to perform qRT-PCR. The total RNA was extracted using Magen (Hi Pure RNA mini Kit) according to the manufacturer’s described protocol. The mixture of primer Script Rt Enzyme (Takara) and oligo (dT) primers was used with RNA. The cDNA was diluted to 100 ng/L. The RT-PCR was run with 30 cycles on a Bio-Rad PTC0200 Peltier Thermal Cycler (Bio-Rad, Hercules, CA, USA). For an internal control, the *EF1α* gene was amplified, and grayscale was used to analyze expression values. The qRT-PCR was performed and, as a template, first-strand cDNA was used with TB SYBR Supermix (Takara). As an endogenous control, *EF1α* was used, and all reactions were carried out three times. To perform qRT-PCR, the following PCR protocol was adopted: initial denaturation at 95 °C for 2 min, followed by 40 cycles of 95 °C for 10 s, and annealing temperature for 30 s. The annealing temperature for *FOXG_04522* was 63.5 °C. To analyze data, the software CFX Manager^TM^ was used. Normalized expression level was determined among mutants and WT using the comparative Ct method (2^−∆∆Ct^), in which ∆∆Ct = (Ct_gene_ − Ct_18srRNA_) mutant − (Ct_gene_ − Ct_18srRNA_) WT. For RT-PCR, specific pairs of gene primers (RT-*Foatg22*F and RT-*Foatg22*R) were designed, as listed in Appendix A). The experiments were performed three times with biological replicates.

### 2.8. Pathogenicity Test

For the tuber inoculation assay, healthy and uniform (100–120 g) potato tubers were used in the present study. Excessive soil and contamination was removed by washing the tubers. For surface sterilization, tubers were dipped for 10 min in a solution of sodium hypochlorite (0.5%) and rinsed with sterile distilled water with three changes, before being cut into small slices and air-dried. The strains of *WT* (*F. oxysporum#1*), *Foatg22∆*, and *Foatg22* overexpression mutant were incubated on potato dextrose agar (PDA) medium plates for 14 days in darkness. From each strain, conidia were collected and washed with pea broth. A 20-µL drop of conidial suspension of 1 × 10^4^ conidia/mL for *WT*, *Foatg22∆*, and *OEFoatg22* was inoculated on potato slices. Five leaves and slices of each potato were placed on moist filter paper in a dish, with three replicates used for each treatment. The inoculated potato slices and leaves were incubated at 27 °C in dark/light conditions for five days. The sizes of each strain on potato slices were measured and analyzed with the Duncan post hoc test. Each experiment was performed twice.

### 2.9. Optical and Epifluorescence Microscopy

To perform epifluorescence and optical microscopic analysis, the aliquots of cells were embedded in 1% agarose blocks and observed under the microscope (M2 Dual Cam) with an appropriate set of filters. The epifluorescence examination was performed using ultraviolet (UV) light (340 to 380 nm) and the following filter blocks: MDC staining (G 365, FT 395, LP 420). To capture the images, a digital camera (EM512 Evolve Photometric) was used with Axiovision 4.8 software. Images were processed using Adobe Photoshop CS3.

## 3. Results

### 3.1. Deletion and Overexpression Mutants of Foatg22 in F. oxysporum

To study the function of *Foatg22*, target gene replacement was performed in the strains of *F. oxysporum* f. sp. *lycopersici*. The hygromycin-resistant (HygR) transformants were analyzed by polymerase chain reaction (PCR) amplification with a specific pair of primers (*Foatg22*F and *Foatg22*R) on insertion flanking regions. The transformants showed an expected shift of the band corresponding to *Foatg22* (Appendix A). These deletion mutants were further confirmed by qRT-PCR with a specific primer pair (RT*Foatg22*F and RT *Foatg22*R) (Figure 1; Appendix A). Overexpression of *Foatg22* was achieved by co-transformation with *Foatg22* genes and different vectors, as confirmed by PCR analysis using specific pairs of primers (*PtrpC*F and *Foatg22*R) (Appendix A). The transformants showed the expected shift band corresponding to *Foatg22* in overexpression (*OE*) mutants (Appendix A). The transformants were further confirmed by qRT-PCR using a specific pair of primers (RT-*Foatg22*F and RT-*Foatg22*R) (Figure 1; Appendix A).

### 3.2. Role of ATG22 in Hyphal Formation

The role of autophagy in several developmental stages of *F. oxysporum* was investigated through observations of autophagosomes. Three-day-old hyphae were collected for staining with monodansylcadaverine (MDC), a dye which is widely used for the detection of the autophagosome in cells [32,33]. In three-day-old hyphae, fluorescence was absent in the deletion mutant compared to wild type (WT), while fluorescence dots of the MDC staining remarkably increased in overexpression mutants (Figure 2). Moreover, we observed that MDC-stained fluorescence dots were similar to overexpression in the wild type (Figure 2). This process of formation of autophagosomes showed that, during the stages of hyphal formation in the wild-type strain and overexpression mutant, the autophagy process was activated. Furthermore, we investigated the role of *ATG22* in phenotype changes in hyphae; the overexpression transformants and deletion mutants were examined for phenotype changes. The hyphal formation in the deletion mutant was significantly different from that in the wild-type and overexpression strains. The wild-type and overexpression strains exhibited dense hyphal formation compared to the deletion mutant (Appendix A). Moreover, after three days of inoculation, hyphal size was not significantly reduced in the deletion mutant compared to the overexpression and wild-type strains (Appendix A). Taken together, the present findings suggest that autophagy contributes to hyphal formation in *F. oxysporum*.

### 3.3. Autophagy Contributes toMycelial Growth and Conidiation

In several fungi, imperfect autophagy severely affects colony growth and conidial formation [33,34,35,36,37]. However, to investigate the role of autophagy in the growth and developmental process of *F. oxysporum*on different media, we adopted overexpression and knockout approaches. Firstly, we evaluated mycelial growth on potato dextrose agar (PDA). The mycelium of *F. oxysporum* mutants *∆Foatg22* and *OEFoatg22* were inoculated in the center of plate and then allowed for grow, while the growth of mycelium was observed over a specific period of time. After two weeks of inoculation, the mycelium of the overexpression (OE) mutant covered approximately the whole surface of the plate, while in the deletion mutant, radial growth was suppressed slightly compared to the wild type (Figure 4A). However, in the deletion mutant, mycelial growth was significantly suppressed and colony diameter was reduced compare to wild type (WT) and the overexpression (OE) mutant (Figure 3B). This finding suggests that autophagy contributes to the mycelial growth of fungus *F. oxysporum.* Moreover, those strains which were grown under nutrient-limited conditions (SM diluted 1:1000) exhibited a faint mycelium with an increase in diameter, but aerial hyphae were not detectable, probably due to severe nutrient-limited conditions (Appendix A). However, the mycelial growth was significantly reduced in deletion mutants compared to wild-type and overexpression strains (Appendix A).

Furthermore, we determined conidial formation in both rich and minimal media. In rich nutrient media (both solid and liquid), after 14 days in solid and after two days in liquid media, the recovered micro conidia were significantly reduced in the deletion mutant compared to the wild type (Figure 3C,D) Likewise, in nutrient-rich media and in minimal media (both solid and liquid), the deletion mutant produced fewer micro conidia compared to the wild-type strain. On the other hand, in overexpression mutants in both conditions, the number of recovered micro conidia was comparable with wild type (Appendix A). Hence, these results revealed that autophagy is important for mycelium growth and conidial formation in *F. oxysporum.*

### 3.4. Expression of Vegetative Growth-Related Gene

To investigate whether *Foatg22* stimulates or inhibits the expression of genes for the production of conidia in *F. oxysporum*, the vegetative growth-related gene was evaluated by qRT-PCR. Firstly, the expression of hydrophobin-encoding *FOXG_04522* was examined, which is an important factor for conidial formation in *F. oxysporum.* As compared with *WT* strains, the gene *FOXG_04522* was not expressed in the deletion mutant, while the expression level of the overexpression mutant was not significantly different from wild type (Figure 4).

### 3.5. Plant Infection Assay

To investigate the role of the deletion mutant in the pathogenicity of *F. oxysporum*, potato (*Solanum tuberosum*) tubers and leaves were used for an infection assay. The potato tubers were inoculated with transgenic strains of *F. oxysporum*. The potato tuber inoculated by wild-type (WT) strains of *F. oxysporum* exhibited progressive symptoms of dry rot disease after seven days of inoculation. Notably, no detectable disease symptoms were observed on those potato tubers which were inoculated with the gene deletion mutant, and symptom development was slightly delayed compared to wild type (WT). The tubers which were inoculated with overexpression (OE) mutants increased the progress of symptom development compared to wild type (WT) and the deletion mutant (Figure 5A,B). These findings suggest that autophagy-related gene *Foatg22* plays a role in the process of infection in *F. oxysporum* during early stages of the disease.

Furthermore, we also used detached potato leaves to test the pathogenicity of deletion mutant of *∆Foatg22*. After seven days of inoculation, the deletion mutant *∆Foatg22* slightly reduced symptom development on potato leaves compared to wild-type and overexpression strains (Figure 5C). Taken together, these results showed that autophagy-related gene *Foatg22* plays an important role in the pathogenicity of *F. oxysporum.*

Moreover, to determine the cause of failure of pathogenicity in the *∆Foatg22* mutant, the infection process was examined microscopically in detail after inoculation. Filamentous fungi start their disease cycle with the germination of conidia. Microscopic analysis showed that, after six days of inoculation, all strains produced conidia spores. The *∆Foatg22* mutant exhibited a low concentration of conidia as compared to the wild type and the overexpression mutant (Figure 6A,B). Inability to produce conidia spores, however, can be an important component in the failure of *∆Foatg22* mutant infectivity.

## 4. Discussion

Previous studies in the filamentous fungi *F. oxysporum* showed that hyphae exhibit an acropetal vegetative growth. Subsequently, *F. oxysporum* can be considered as a mono nucleated compartmented mycelial organism [38]. This proposes the nearness of a system that decisively controls cell volume, compartmentalization, and cell cycle, subsequently keeping up the vegetative development and combination. In the last decade, a number of studies were carried out to explore the role of autophagy-related genes in filamentous fungi *Colletotrichum orbiculare, Fusarium oxysporum*, and *Magnaporthe oryza,* [12,17,18]. However, it is still unknown whether *atg22* participates in autophagosome formation. Here, we found that the deletion mutant of *atg22* suppressed autophagosome formation compared to wild type. On the other hand, in the overexpression mutant, formation of autophagosomes was enhanced compared to the deletion mutant and wild type. These findings revealed that *atg22* may contain an important function to regulate autophagosome formation in *F. oxysporum*. Also, this finding shows a difference from previously reported functions of *atg22* in yeast and *F. oxysporum*, which may be due to diverse functions in different species, and this can provide new insights for further studies [12,13]. The nonappearance of MDC-positive staining in ∆*Foatg22* emphatically proposes that *Foatg22* is a basic component for autophagy in *F. oxysporum*.

Endogenous recycling of the cytosol and organelles via autophagy is recommended to be vital for nutrient transport along hyphal filaments, as well as for the improvement of aerial hyphae which contain conidiophores [39,40]. In line with this thought, *∆Foatg22* strains displayed diminished aerial hyphae and conidiation, as reported in different filamentous fungi [33,34,35,36,37]. In virulent pathogens, successful virulence might rely upon the reusing of macromolecules to help cell movement under nutrient-restricting conditions. Filamentous fungi start their disease cycle with the germination of conidia. In *M. oryzae*, loss *of MoATG8* resulted in autophagic conidial cell demise, prompting hindered appressorium arrangement, loss of pathogenicity, and decreased conidiation [16]. As shown in its capacity in *M. oryzae*, *CoATG8* is moreover associated with conidiation, appressorium arrangement, and pathogenicity in *C. orbiculare* [17].

In a deletion mutant of *aoatg8*, conidial formation was delayed, which revealed that, in filamentous fungi, autophagy is required during early stages of conidial germination [36]. In agreement with previous studies, we found that the virulence of ∆*Foatg22* strains on both potato [12] tubers and on detached leaves was significantly attenuated, producing fewer conidia spores compared to wild type and overexpression mutant; however, this can be an important component in the failure of the *∆Foatg22* mutant.

The genes *VdPKAC1* and *VdSNF1* are the known responsible factors for virulence in eggplant and tomato, while *VMK1* is also responsible for virulence on both tomato and tobacco, and *VdNLP1* and *VdNLP*2 are important factors in the virulence of cotton, *Arabidopsis*, and tobacco [10,11,25,26,27,41]. Autophagy plays an important role in the vegetative growth of *F. oxysporum* [11]. In the present study, the gene *FOXG_04522*, which is a well-known vegetative growth-related gene of *F. oxysporum*, was used. The deletion mutant of *Foatg22* inhibited the expression level of pathogenicity-related genes during conidial formation. However, gene expression was consistent with phenotype. Finally, these findings verified the unreported assumption that *atg22* is essential for the formation of autophagosomes during the vegetative growth of *F. oxysporum*. Our findings strongly recommend that autophagy influences *F. oxysporum* to regulate vegetative growth. However, further studies are required to explore the exact role of *atg22* during vegetative growth and the formation of autophagosomes.

## Figures and Tables

**Figure 1 genes-10-00365-f001:**
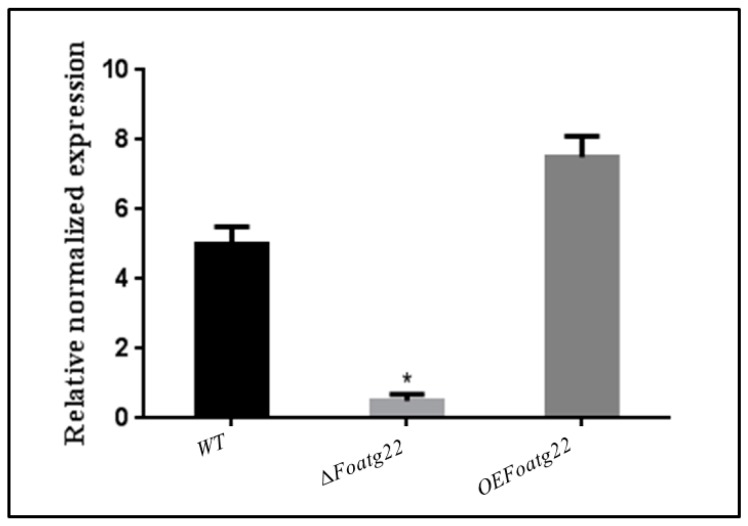
Quantitative real-time reverse-transcription polymerase chain reaction (qRT-PCR) analysis of wild-type (WT), ∆*Foatg22*, and overexpression strains. Three biological replicates were used for this study. The error bars show standard deviation. To perform statistical analysis, the Duncan post hoc test was used (*p* < 0.05).

**Figure 2 genes-10-00365-f002:**
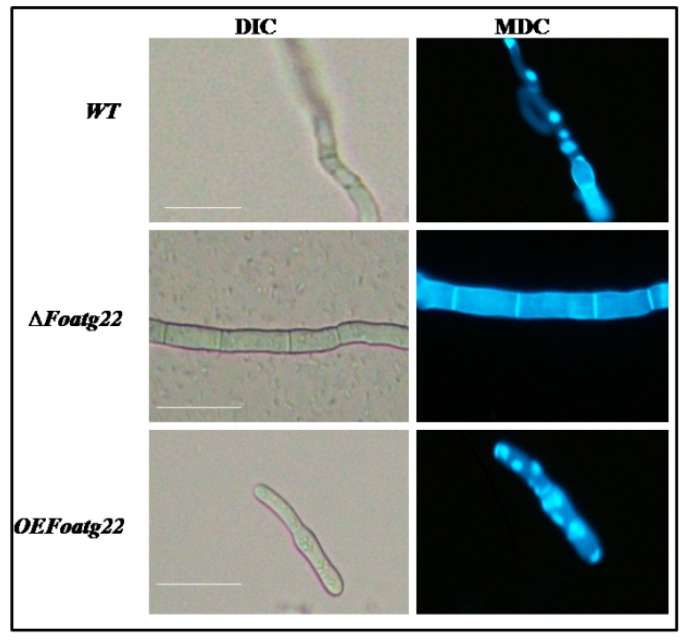
Autophagy formation in three-day-old hyphae of strains which were stained with monodansylcadaverine (MDC). Fluorescence dots show the formation of autophagosomes. Bars = 15 µm.

**Figure 3 genes-10-00365-f003:**
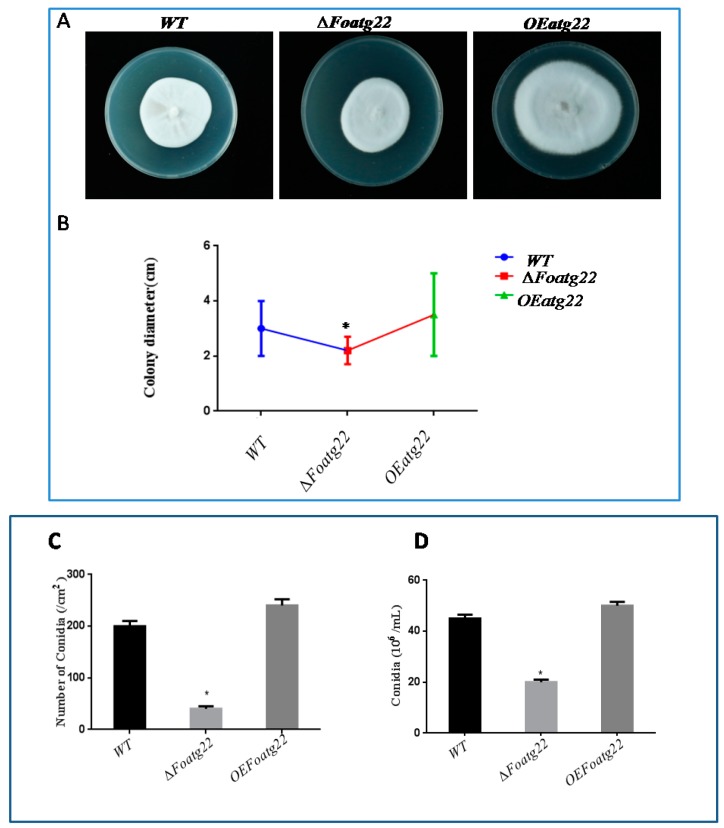
In ∆*Foatg22*mutants, conidial formation and hyphal formation was reduced. (**A**) Image of 14-day-old strains after inoculation. (**B)** Fresh micro conidia (10^3^) were inoculated on potato dextrose agar (PDA) plates and then incubated at 28 °C. Colony diameter was measured on a daily basis until two weeks. The graph represents the growth rate of strains. (**C**)Number of micro conidia recovered from PDA plates grown after 14 days, at 28 °C. Conidial formation was significantly reduced in *∆Foatg22*. (**D**) The number of recovered micro conidia from a two-day-old culture with shaking at 28 °C. Conidial formation was reduced significantly in *∆Foatg22*. Experiments were performed three times with similar results. Bars indicate standard error from replications. To perform statistical analysis, the Duncan post hoc test was used (*p* < 0.05).

**Figure 4 genes-10-00365-f004:**
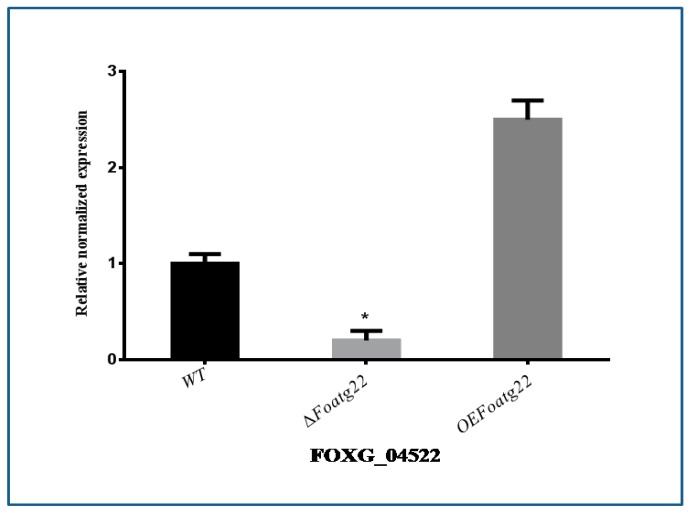
Verification of mutants by qRT-PCR. Quantitative real-time reverse-transcription polymerase chain reaction (qRT-PCR) analysis of vegetative growth-related gene in wild-type (WT), ∆*Foatg22*, and overexpression strains. Three biological replicates were used for this study. Statistical analysis was performed using the Duncan post hoc test (*p* < 0.05).

**Figure 5 genes-10-00365-f005:**
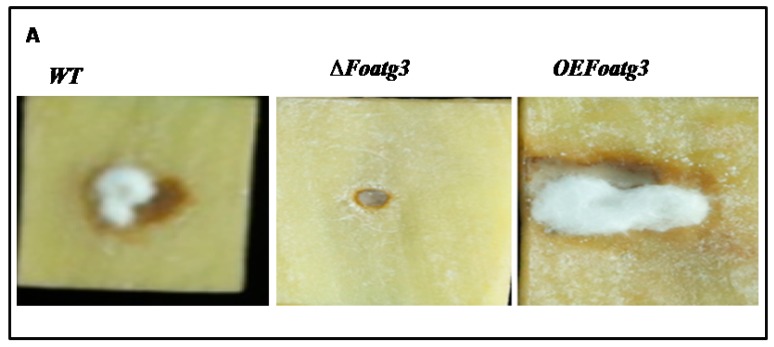
*Foatg22* contributes to virulence on plant. (**A**) Disease progress in the tubers which were inoculated with wild type (*WT*)*, ∆Foatg22,* and overexpression (*OE*) *atg22*. (**B**) Potato tubers were inoculated with a suspension of 5 × 10^6^ freshly obtained micro conidia per mL of wild-type (WT), ∆*Foatg22*, and *OE Foatg22* strains. Percentage of symptoms was recorded after seven days. (**C**) Potato leaves for each treatment were inoculated with a suspension of 5 × 10^6^ freshly obtained micro conidia per mL of wild-type (WT), ∆ *Foatg22*, and *OEFoatg22* strains. Three replicates were used for all experiments with similar results (Duncan, *p* < 0.05).

**Figure 6 genes-10-00365-f006:**
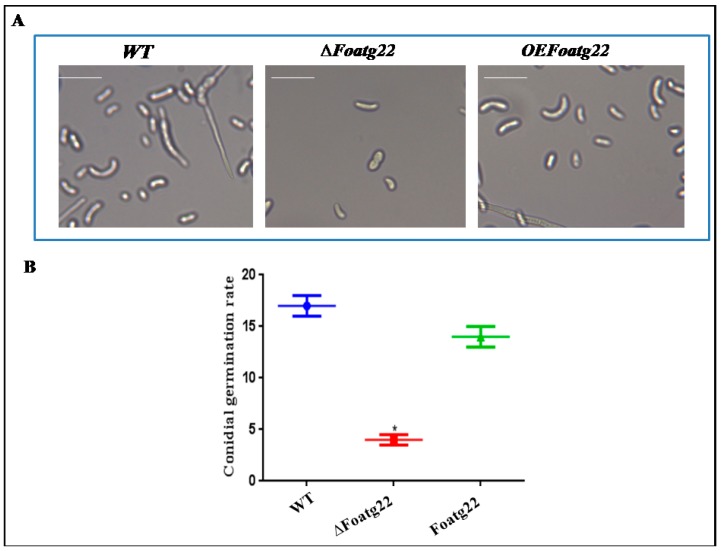
Hyphal germination was reduced in ∆*Foatg22* mutants: (**A**) conidial germination; (**B**) conidial germination rate. Bars indicate the standard error from three independent replications (Duncan < 0.05); Bars = 10 µm.

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
