# Peer review of "Role of Autophagy-Related Gene atg22 in Developmental Process and Virulence of Fusarium oxysporum"

_genes, 2019, doi:10.3390/genes10050365_

Round 1

Reviewer 1 Report

Type of Manuscript: Article

Title: Role of autophagy related gene ATG22 in developmental process and virulence of F. oxysporum

Authors: A. Rehman Khalid, Shumin Zhang, Xiumei Luo, Khalid Mehmood, 
Hamayun Shaheen, Junaid Rahim, Pan Dong, Dan Qui, Maozhi Ren

Journal: Genes

The manuscript describes the effect of the deletion and overexpression of a gene involved in autophagy on virulence and development. Data shown are clear, however, the text describing the results, the methods, and the discussion are not easily understood. Sometimes is due to spelling mistakes or due to the use of the language, but sometimes is due to lack of information or lack of words. English of the manuscript should be extensively reviewed and corrected. I have suggested many changes, but I haven't included all of them. There are many words starting with capital letters that should not be. I include my comments with some changes and recommendations to improve the quality of the manuscript. The text should be corrected and all the experimental details should be well described.

Title

In the title, Fusarium should be written complete and not abbreviated (F. oxysporum).  The name of the gene should be written as Foatg22.

Introduction

Names of genes for Fusarium and other fungi are written in lower case. Genetic nomenclature has to be corrected. As an exception, yeast genes are written in capital letters. I recommend to rename the gene (Foatg22), as well as, the name of the deletion mutant (DFoatg22 or Foatg22D). A similar gene and deletion mutants from F. oxysporum, Foatg8D, has been published. Nomenclature can be checked in Corral-Ramos et al (2015).

Material and Methods is generally included after introduction in journal Genes. It is also useful to see the figure describing how the construction for the deletion and overproduction of the gene was carried out before reading the results.

Line 243: How many strains were used in this study? And what is the name of the strains? According to the results, only 1 transformant from 1 strain is described. Usually a number collection is given for the strain used in a study to distinguish from other strains of Fusarium oxysporum.

I suggest that the generation of deletion mutants (now in 4.4) appears before fungal transformation (4.2).

Line 284, Describe what U-Hph-D is in the text as it appears in a supplementary figure. Does it mean Upstream-Hph-Downstream? Hph should be hph as it is the name of a gene.

Line 288, write the complete name for F2du (5-fluoro-2'-deoxyuridine). It is not understood for everybody.

Line 353, Check the volume of the drop, was it 20 microliters?

Line 353, the term sporangial and sporangia is not correct. Sporangia is the name of the receptacle where spores are formed. The correct term is spores, conidia or microconidia.

RESULTS

Line 69: Rewrite

Line 71: Include name of primers used for screening of transformants.

Line 72: insert "band corresponding to the Foatg22"

Line 73: Include name of primers used for pRT-PCR

Idem for overexpression mutants

Figure 1. There is no indication of the name of the genes whose expression was analyzed.

Nomenclature of the deletion and overexpression mutants should be changed

Line 84, Replace "was" with "were"

Line 92: Correct spelling of "transforments". It should be transformants

Line 113: More over should be "moreover"

Line 118: Correct spelling of sever, should say severe

Line 125: There is a repetition Wild type and WT

Line 126: Insert "the number of" between "conditions" and "recovered micro conidia"

Line 134: Correct verb, it should say: "represents" instead of "represent"

Line 134: Change the order of the sentence: Number of micro conidia recovered from PDA plates grown after 14 days, at 28ºC.

Page 6: Figure 3 C, If the differences observed in the colony diameter were significant, the number of conidia should be counted by cm2 instead of counting them per plate. In any case, I expect that the number of conidia will be significantly different.

Fig 3 D: In the case of the conidia counted in liquid media, number of conidia should be referred to biomass. Is the deletion mutant growing less than WT in liquid cultures? This information should be included.

Line 140, Substitute period (.) by coma (,) and substitute period "the" by coma "a"

Line 141, avoid use of "we". Use impersonal tense: Expression of FOXG_04522 was examined. Include the name of the protein coded by that gene.

Line 142, correct verb tense: the gene....was not expressed in the deletion mutant

148, Passive tense is suggested to be used: "Statistical analysis was performed using ...."

Renumber figures, there are two figures number 4

Line 170, indicate the number of spores used in the virulence assays instead of the concentration.

Line 178, correct verb tense: it should say "to produce"

Figure 5: Germlings are out of focus. These photos should be repeated.

Line 194, Spelling of mechanism

Line 203, Check spelling of decay, do you mean "decade"?

Line 234, Insert "a", It should say "a well-known vegetative"

References

Revise and correct the references to follow style of Genes

- In many references, latin names of species are not written in italics, revise and correct.

- Check that name of journals are abbreviated

- Check the title of articles, they should be in sentence style.

....

Supplementary figures

Figure S1: Label U and D in the WT locus and replacement construct in Figure S1.

Locus is more correct than loci, as loci is the plural and the figure refers to only 1 gene.

Loci should be replaced by locus in "Mutant loci" and "Wt loci"

Figure S2, legend of Figure S2 should be improved. Remove "Fig.1". Legend of Figure S2 should indicate that is a PCR of genomic DNA and it should describe the primers used for the amplifications. Moreover, at least some of the sizes of the bands of the DNA marker should be indicated.

Figure S4

S4. A) I cannot see the colonies on the plates, I recommend to remove because don't give any information.

S4 C. If there are differences in the area of the colonies, the number of conidia should be calculated per area, for instance, number of conidia per cm2.

S4 D) Number of conidia should be referred to biomass. Is the deletion mutant growing less than WT in liquid cultures? This information should be included.

Author Response

Dear Reviewer ,

We would like to thank the reviewer for careful and thorough reading of this manuscript and for the thoughtful comments and constructive suggestions, which help to improve the quality of this manuscript. Our response follows (the reviewer’s comments are in italics). Please find an attachment.

Reviewer 2 Report

File attached

Author Response

Dear Reviewer, 

We would like to thank the reviewer for careful and thorough reading of this manuscript and for the thoughtful comments and constructive suggestions, which help to improve the quality of this manuscript. Our response follows (the reviewer’s comments are in italics). Please find an attachment.

Round 2

Reviewer 2 Report

Attached

Author Response

We would like to thank the reviewer for careful and thorough reading of this manuscript and for the thoughtful comments and constructive suggestions, which help to improve the quality of this manuscript. Our response follows (the reviewer’s comments are in italics).
